

# Phenomenological extraction of a universal TMD fragmentation function from single hadron production in $e^+e^-$ annihilations

**Mariaelena Boglione**[1,2⋆], **J.O. Gonzalez-Hernandez**[1,2] and **A. Simonelli** [1,2]

**1** Physics Department, University of Turin
**2** INFN - Sezione Torino

⋆ boglione@to.infn.it

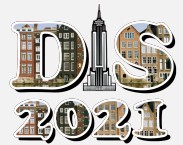

## Abstract

**Factorizing the cross section for single hadron production in $e^+e^-$ annihilations, differential in $z_h$, $P_T$ and thrust, is a highly non trivial task. We have devised a factorization scheme that allows us to recast the $e^+e^- \to hX$ cross section in the convolution of a calculable hard coefficient and a universal Transverse Momentum Dependent (TMD) Fragmentation Function (FF). The predictions obtained from our NLO-LL perturbative computation, together with a simple ansatz to model the non-perturbative part of the TMD, are applied to the experimental measurements of the BELLE Collaboration for the phenomenological extraction of this process independent TMD FF.**

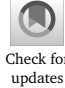

## 1 Introduction

TMD phenomenological studies have traditionally been carried out mainly exploiting three hadronic processes, where factorization has been proven to hold: Drell-Yan scattering, $e^+e^-$ annihilations in two almost back-to-back hadrons and Semi-inclusive Deep Inelastic Scattering (SIDIS), for which we have a reasonably large amount of experimental data available. In all of the above processes two hadrons are involved, for this reason we classify them as belonging to the 2-hadron class. This influences the mechanism of factorization of the corresponding cross section, especially affecting the way in which collinear and soft parts are separated and the subtraction mechanism, the treatment of rapidity divergences and, ultimately, the definition of the TMDs themselves.

Regardless of all the details of the factorization procedure, for which we refer to Refs. [1–4] even a very superficial inspection of the 2h-class cross section will show that it inevitably contains the convolution of two parton densities, each associated to one of the two involved

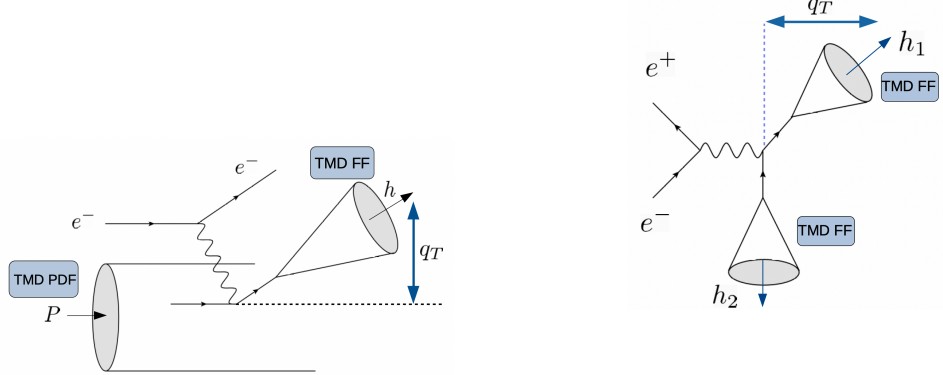

Figure 1: Pictorial representation of a SIDIS process (left panel) and an $e^+e^- \to h_1 h_2 X$ scattering (right panel)

hadrons: Drell-Yan cross sections will be proportional to the convolution of two Parton Distribution Functions (PDFs), for $e^+e^- \to h_1 h_2 X$ we will have two fragmentation functions (FFs), while SIDIS will see one PDF and one FF, as shown in Fig. 1. Although in principle one should be able to extract these functions by simultaneously fitting experimental data from the above processes, in practice this turns out to be a very complicated challenge, as the distribution and/or fragmentation functions appear in the cross section as a convolution, which implies a strong entanglement of the two functions.

One additional difficulty is related to the fact that hard, soft and collinear contributions to the cross sections are defined in the impact parameter space, the so called $b_T$ space, while experiments naturally deliver information on the transverse momenta of the hadrons (and related partons) involved in the process. The cross sections can be written as follows

$$\frac{d\sigma}{dq_T} = \mathcal{H}_{\text{SIDIS}} \int \frac{d^2\vec{b}_T}{(2\pi)^2} e^{i\vec{q}_T \cdot \vec{b}_T} F(b_T) D(b_T), \tag{1}$$

$$\frac{d\sigma}{dq_T} = \mathcal{H}_{\text{2-h}} \int \frac{d^2\vec{b}_T}{(2\pi)^2} e^{i\vec{q}_T \cdot \vec{b}_T} D_1(b_T) D_2(b_T), \tag{2}$$

where $\mathcal{H}_{\text{SIDIS}}$ and $\mathcal{H}_{\text{2-h}}$ are the hard factors associated to the partonic process, while $F(b_T)$ and $D_{(1,2)}(b_T)$ are the TMD distribtion and fragmentation functions related to the collinear factor associated to each hadron involved in the scattering process, appropriately subtracted by (a proper combination of) the corresponding soft factors [1–4]. While $F$ offers a 3D-picture of partons inside the target hadron, $D$ sheds light on the mechanism of hadronization of partons into hadrons. However, these functions are of no practical use unless they are Fourier transformed to the $k_T$ momentum space, so that the $b_T$ dependent cross section resulting from the factorization procedure is recast into the corresponding $q_T$-dependent, observable cross sections that can be experimentally accessed. This clearly makes any phenomenological analysis rather cumbersome.

One way to circumvent the problem of simultaneously extracting two different, entangled functions from the same (observable) cross section is to consider processes in which only one hadron is involved. According to Ref. [4] these processes are assigned to the 1h-class. In 2019, the BELLE collaboration at KEK published their data on this cross section, with the transverse momentum of the observed hadron measured with respect to the thrust axis [5]. This is one of the measurements which go closer to being a direct observation of a partonic variable, the

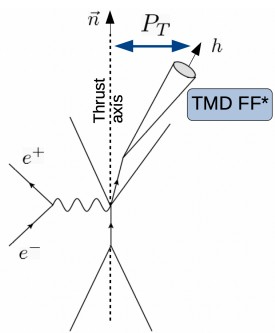

Figure 2: Pictorial representation of an $e^+e^- \to hX$ scattering process.

transverse momentum of the hadron with respect to its parent fragmenting parton, which has indeed triggered a great interest of the high energy physics community, especially among the experts in the phenomenological study of TMD phenomena and factorization. A pictorial representation of this process is given in Fig. 2

With its cross section depending on only one TMD, the $e^+e^- \to hX$ process offers an ideal framework to access the unpolarized TMD fragmentation function

$$\frac{d\sigma}{dP_T} = d\widehat{\sigma} \otimes D^\star(P_T).\tag{3}$$

However, one should be very careful to the deceitful pitfalls of a simplistic reasoning. In fact, the TMD fragmentation function appearing in the $e^+e^- \to hX$ cross section, $D^\star(b_T)$ has very little in common with the TMDs appearing in $e^+e^- \to h_1 h_2 X$, the main difference originating in the corresponding gluon soft factors. While in a 2h-class process this non perturbative contribution is evenly shared by the two TMDs associated to $h_1$ and $h_2$, thanks to the so-called "square root definition" [1,4], in $e^+e^- \to hX$ the gluon soft factor turns out to be a perturbative (computable) contribution, corresponding to the soft thrust function in the partonic cross section. Consequently $D^\star(P_T)$ is defined as a purely collinear object, totally free from any soft gluon contribution, very differently from the 2-h class TMD. We indicate this new TMD definition by "factorization definition".

In Ref. [4] we showed that a relation exists between $D(P_T)$ and $D^\star(P_T)$, that can be written as follows:

$$D(b_T) = D^\star(b_T)\sqrt{M_S}(b_T)\tag{4}$$

where $M_S(b_T)$ is the soft model, which represents the long distance, fully non-perturbative behaviour of the TMD fragmentation function. The advantage of using this definition, with the soft gluonic contribution completely separated out from the the collinear part, is that now $D^\star$ is universal, i.e. it is the same for all processes belonging to any hadron class. Consequently, $D^\star$ can safely be extracted from Drell-Yan,SIDIS as well as from $e^+e^- \to hX$ (even from simultaneous fits). The non-universal, process-dependent part, has now been singled out and is entirely contained in the soft model function $M_S$ which, instead, will have to be extracted separately for each hadron class.

## 2 Factorization of the $e^+e^- \to hX$ cross section to NLO-LL accuracy

The factorized $e^+e^- \to hX$ cross section is differential in three variables:

- The fractional energy $z$ of the detected hadron,

- The thrust $T$, defined as

$$T = \frac{\sum_i |\vec{P}_{(\text{c.m.}),i} \cdot \widehat{n}_{\text{had.}}|}{\sum_i |\vec{P}_{(\text{c.m.}),i}|}. \tag{5}$$

- The transverse momentum $P_T$ of the detected hadron with respect to the thrust axis, assumed to be the direction of the jet to which $h$ belongs, the same direction of the fragmenting parton.

The measurement of thrust is crucial to obtain a TMD observable from $e^+e^-$ annihilation into a single hadron. In fact, the thrust axis provides a valid estimate of the axis of the jet in which the hadron is detected, which coincides with the direction of the fragmenting parton, i.e. the reference direction with respect to which the transverse momentum $P_T$ of the detected hadron is measured. Most importantly, the value of $T$ also introduces a further correlation among the momenta generating the soft and the collinear subgraphs, in addition to the usual correlation induced by momentum conservation. Clearly the partonic cross section will depend on $T$ but, as explained in Ref. [4], also the TMD FF acquires a dependence on thrust, through its rapidity cut-off $\zeta$. Notice that, crucially, TMDs are not invariant with respect to the choice of $\zeta$ (see Refs. [3,4] and references therein for a very detailed discussion). The $e^+e^- \to hX$ cross section is then written as

$$\frac{d\sigma}{dz_h \, dT \, dP_T^2} = \pi \sum_f \int_{z_h}^1 \frac{dz}{z} \frac{d\widehat{\sigma}_f}{dz_h/z \, dT} D_{1,\pi^\pm/f}(z, P_T, Q, (1-T)Q^2). \tag{6}$$

The relation between $\zeta$ and $T$ can be made explicit by introducing a **topology cut-off**, $T_{min}$, that forces the partonic cross section to describe the proper final state configuration, i.e. a pencil-like configuration, which is obtained in the 2-jet limit, $T \to 1$. Large $T_{min}$ will correspond to large rapidities, i.e. a very well collimated jet, as represented in Fig. 3. In practice, the topology cut off can easily be related to the power counting, IR-scale $\lambda$, which sets the upper limit in the $k_T$ integration. Summarizing, in a very schematic way:

$$T > T_{min} \implies 2-\text{jet limit} : T_{min} \to 1, \ \lambda \to 0 \tag{7}$$

$$k_T \leq \lambda, \ k_T \leq \sqrt{\tau} Q \implies \lambda = \sqrt{\tau} Q. \tag{8}$$

Having exploited the measured value of the hadronic thrust to fix the rapidity cut off, the final results for the partonic cross section and the TMD can be written as

$$\frac{d\widehat{\sigma}_f}{dz \, dT} = \left[ -\sigma_B e_f^2 N_C \frac{\alpha_S(Q)}{4\pi} C_F \delta(1-z) \left[ \frac{3 + 8\log\tau}{\tau} \right] + \mathcal{O}\left(\alpha_S(Q)^2\right) \right] e^{-\frac{\alpha_S(Q)}{4\pi} 3C_F (\log\tau)^2 + \mathcal{O}\left(\alpha_S(Q)^2\right)} \tag{9}$$

$$\begin{aligned} \widetilde{D}_{1,\pi^\pm/f}(z, b_T; Q, \tau Q^2) = \frac{1}{z^2} \sum_k &\left[ d_{\pi^\pm/k} \otimes \mathcal{C}_{k/f} \right](\mu_b) \times \\ &\times \exp\left\{ \frac{1}{4} \widetilde{K} \log\frac{\tau Q^2}{\mu_b^2} + \int_{\mu_b}^Q \frac{d\mu'}{\mu'} \left[ \gamma_D - \frac{1}{4} \gamma_K \log\frac{\tau Q^2}{\mu'^2} \right] \right\} \times \\ &\times (M_D)_{f,\pi^\pm}(z, b_T) \exp\left\{ -\frac{1}{4} g_K(b_T) \log\left( \tau \frac{Q^2}{M_H^2} \right) \right\} \end{aligned} \tag{10}$$

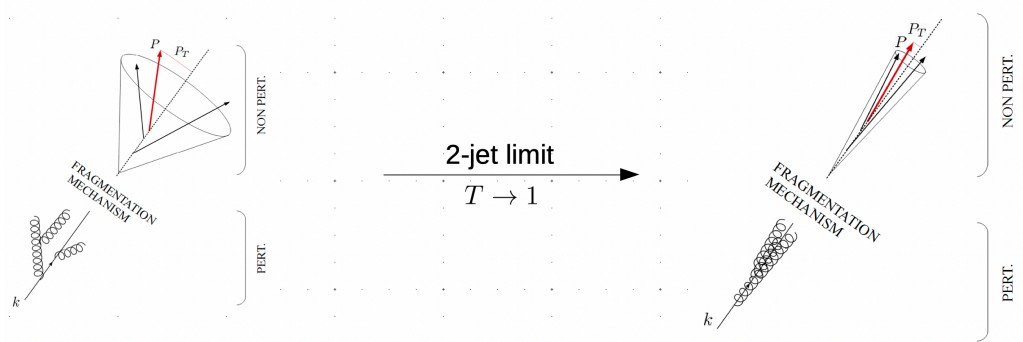

Figure 3: Pictorial representation of a TMD Fragmentation Function, in which the separation between perturbative and non-perturbative regime is explicitly shown, corresponding to two different values of the rapidity cut-off. On the right panel a very large value of the rapidity cut-off reproduces a 2-jet limit configuration.

where $\tau = 1 - T$, $M_D(z, b_T)$ embeds the non-perturbative, long-range behavior of the TMD FF and $g_K(b_T)$ is a universal function, independent of the TMD definition used. Notice that the first two lines of Eq. (10) correspond to the perturbative content of the TMD, while the last line represents its non-perturbative part, which will have to be modelled and extracted from experimental data.

## 3 Preliminary phenomenological results

In this last Section we will present some preliminary results of our phenomenological analysis of the $e^+ e^= \to hX$ cross section as measured by the BELLE experiment [5].

To model the non-perturbative part of the TMD FF we will exploit a parameterization that, in the $b_T$ space, corresponds to a Bessel-K function, normalized in such a way that it is 1 at $b_T = 0$:

$$(M_D)_{f, \pi^\pm}(z, b_T) = z^{-\rho b_T^2} \frac{2^{2-p}}{\Gamma(p-1)} (b_T m)^{p-1} K_{p-1}(b_T m) \tag{11}$$

where $K_{p=1}$ is the modified Bessel function of the second kind. This model was successfully applied in Ref. [6] and, more recently, in Ref. [3, 4]. It is known as the "power law model", since its Fourier transform is closely reminiscent of the propagator of an on-shell fermion (see dedicated discussion in Ref. [7]). Notice, however, that here we introduce an additional $z$ dependence with a Gaussian $b_T$ functional shape, namely the factor $z^{-\rho b_T^2}$, that allows us to describe the $z$-dependence of the cross section remarkably well, for all the $z$-bins covering the range $0.2 < z < 0.8$, as shown in Fig. 4. The fit includes data with $P_T \le 0.2 z_h Q$, to make sure we remain in the TMD region where our model is expected to work, for a total number of 89

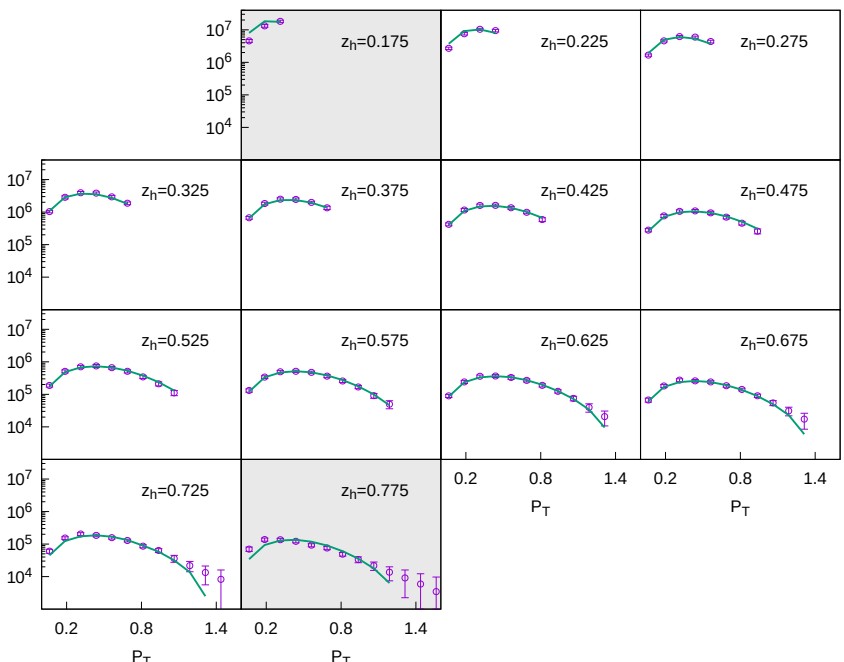

Figure 4: Cross section as a function of $P_T$ for different bins in $z$, at fixed $T = 0.875$. Data in the shaded boxes are not included in the fit.

fitted points. The total $\chi^2_{dof}$ of the fit is 0.9.

Finally, we also test the predictive power of Eqs. (9) and (10) to reproduce the correct $T$-dependence. This is, at the moment, the most delicate issue of this phenomenological analysis. which is still under scrutiny. Here we only point out the very subtle interplay between $z$ and $T$ which is mainly controlled by the shape of the non-perturbative $g_K(b_T)$ function, which is commonly (but not necessarily) modeled as a Gaussian in $b_T$. Fig. 5 shows some very preliminary results, corresponding to a restricted selection of $z$-bins, approximately $0.4 < z < 0.6$, for two values of thrust, $T = 0.825$ (left panel) and $T = 0.825$ (right panel).

## 4 Conclusion

In this paper we have presented some preliminary results of the phenomenological analysis of BELLE experimental data [5], differential in $z$ and $P_T$ as well as in thrust, $T$. Although an analysis of the same data, based on a Soft Collinear Effective Theory, was released in Ref. [8], we have been able, for the first time, to describe the thrust dependence of the $e^+e^- \rightarrow hX$ cross section.

Future plans, after completion of the present study, include the extraction of the Soft Model $M_S(z, b_T)$, obtained from the ratio $M_S = D/D^\star$, see Eq. (4) , from a combined analysis of SIDIS and $e^+e^-$ data. Finally, we will be able to determine the TMD PDF, $F^\star$ from a global simultaneous fit of Drell-Yan, $e^+e^-$ and SIDIS data. This will be possible thanks to the formalism proposed in Refs. [3,4], that allows to define the TMDs in a completely universal way, separating out the soft (process dependent) part from the collinear(process independent) part, finally restoring the possibility to perform global analyses including data from processes belonging to different hadron-classes.

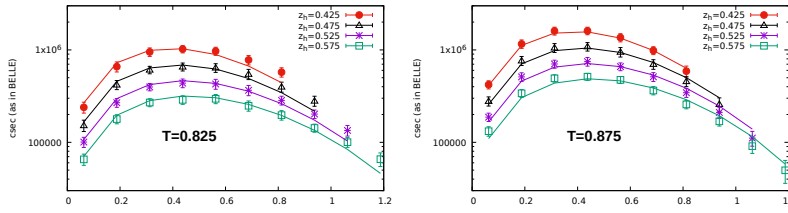

Figure 5: Cross section as a function of $P_T$ for two different bins in thrust, $T = 0.825$ (left panel) and $T = 0.825$ (right panel), corresponding to a restricted selection of $z$-bins, $0.4 < z < 0.6$.

# Acknowledgements

We are grateful to the organizers for giving us the possibility to present our work at DIS 2021, despite the very difficult situation due to the Covid-19 pandemics.

**Funding information** This project has received partial funding from the European Union's Horizon 2020 research and innovation programme under grant agreement No 824093.

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
