# Peer review of "Phenomenological extraction of a universal TMD fragmentation function from single hadron production in $e^+ e^-$ annihilations"

_SciPost Physics Proceedings, doi:SciPost Phys. Proc. 8, 139 (2022)_

## Round 1 · Referee Report · Anonymous (Referee 1) · 2022-3-1

Report

For those less closely familiar with the field, it would be appreciated to clarify what is meant by 'only one hadron involved' in the context of e+e- --> dijet events, which involve multiple final-state hadrons. The Journal's acceptance criteria for these proceedings are otherwise met.
  • validity: -
  • significance: -
  • originality: -
  • clarity: -
  • formatting: -
  • grammar: -

Author:  Mariaelena Boglione  on 2022-03-02  [id 2258]

(in reply to Report 1 on 2022-03-01)

For the process e+e- ---> hadron + X , BELLE has measured the cross section for two-hadron production as well as one-hadron production, namely e+e- ---> h1 + h2 + X as well as e+e- ---> h + X. For the first process usual TMD factorization theorems can be applied, while for the second process the factorization properties are still under debate. In this paper we focus on the analysis of the latter process, e+e- ---> h + X.
I think the introduction specifies this point well enough.

Anonymous on 2022-03-03  [id 2265]

(in reply to Mariaelena Boglione on 2022-03-02 [id 2258])
Category:
remark

Thank you for the reply. Although the overall subject is clear, it is not as clear whether the discussion is always with regards to the entire h+X final state, or some subset. The requested revision, that the author clarify the meaning of the 'one hadron involved', was meant to address this: While a theorist may understand this to mean 'only one hadron meaningfully present in the calculations', an experimentalist might well take it as 'only one hadron present in the final state', which is clearly not the case in the context of a jet.

Taken as a whole, the paper does not support the second interpretation, and a sufficiently invested reader will come to this conclusion even if the misleading wording remains. Since the author believes this does not merit revision, this reviewer relents.

---

## Editorial Decision

published